# Accuracy and real time optimization of remote sensing image change detection based on IRAU and DSC

**Yingying Liu** [ID]*

School of Information Engineering, Henan University of Animal Husbandry and Economy, Zhengzhou, China

* liuying85815@163.com

## Abstract

Image change detection is one of the important application branches of remote sensing technology in many fields. However, in complex environments, remote sensing image change detection is often subject to various interferences, resulting in low accuracy and poor real-time performance of detection results. The research focuses on the advantages and problems of residual networks and depth-wise separable convolution modules, designs a new remote sensing image change detection model, and finally sets up experiments for verification. The average accuracy of the proposed detection model before and after training convergence was 0.54 and 0.97. The accuracy of repeated detection ranged from 95.82% to 99.68%, and the area under curve of the model was 0.90. However, after removing the integrated residual attention unit and depth-wise separable convolution, the accuracy decreased by 1.91% and the latency increased by 117ms. In addition, the detection efficiency of the model for different elements ranged from 0.91 to 0.94, with high accuracy in detecting changes in spatial and temporal scales and small offsets. The actual accuracy and mean latency time of the model were 92.43% and 260ms, respectively. In summary, the proposed change detection model significantly improves the accuracy and real-time performance of remote sensing image processing, contributing to the expanded application of remote sensing dynamic detection technology in fields such as ocean monitoring and ecological research.

## 1. Introduction

Driven by technological development, the population has rapidly expanded, and the acquisition and processing of information at the geographic spatial scale have received great attention [1]. Remote sensing technology has wide coverage, fast data acquisition speed, large amount of information, and is not limited by ground conditions. It is widely used in environmental investigation and assessment, urban

**Data availability statement:** All relevant data are within the paper and its Supporting Information files.

**Funding:** The author(s) received no specific funding for this work.

**Competing interests:** The authors have declared that no competing interests exist.

planning and management, which is also a research focus in computer vision [2–3]. Remote Sensing Image Change (RSIC) detection is a crucial application direction of remote sensing technology. According to the differences in detection ideas, there are pixel change detection and object change detection [4–5]. However, due to the increasing complexity of the Earth's environment, traditional RSIC detection approaches have been unable to meet practical needs. For example, change detection methods based on object-oriented image analysis heavily rely on the segmentation results of image targets [6]. The automatic classification accuracy of time series change detection methods is extremely low and only used for large-scale target change analysis [7]. The three-dimensional change detection method requires high accuracy of the three-dimensional data itself and is computationally complex [8]. These issues seriously affect the accuracy and real-time performance of RSIC detection. Therefore, the research focuses on optimizing the effective acquisition and accurate detection of remote sensing images, focusing on the basic logic and excellent performance of modules such as Residual Networks (ResNets) and Depth-wise Separable Convolution (DSC), and making improvements. An Integrated Residual Attention Unit (IRAU) is proposed. Finally, a RSIC detection model on the basis of IRAU and DSC is constructed. The research aims to optimize the accuracy and real-time performance of RSIC detection models, enhance their resistance to environmental interference, and make positive contributions to the wider application, laying the foundation for promoting the rational planning and development of geographic spatial scales. The innovation lies in the organic combination of ResNet, attention mechanism and other modules. The IRAU unit is proposed and combined with DSC for improvement, establishing a new detection network architecture that effectively improves the detection accuracy and reduces detection latency.

The research is structured from four parts. Part one introduces the current research on RSIC detection worldwide. The second part starts from modules such as ResNets, Split and Concat (SPC), and DSC to establish an accurate and real-time RSIC detection model. The third part provides numerical examples and practical application analysis of the proposed image detection algorithm model to verify its reliability. The final section provides a comprehensive summary and analysis of the article.

## 2. Related works

With population growth and the large-scale development of various industries, the application of remote sensing technology in natural resources, agricultural production and other fields is rapidly increasing [9–10]. RSIC detection is the core of achieving geographic scale monitoring, which is also a crucial application direction that remote sensing technology needs to continuously expand and deepen [11]. However, in practical work, the performance of RSIC detection in complex environments is not stable, so many researchers are improving this problem [12]. C. Zhang et al. built a self-attention network on the basis of Siamese U-shaped structure to address the convolution operations being unable to capture global information, effectively improving the efficiency of processing detail information in RSIC [13]. G. Cheng et al. built

an optimized separability deep learning network to address the boundary blurring between different semantic hierarchical features caused by traditional backbone networks. Through boundary maximization and directed attention mechanisms, the accuracy of RSIC detection was improved [14]. In response to the misalignment problem in traditional backbone networks, Y. Feng et al. designed an intra scale cross interaction and inter scale feature fusion network, which enhanced the integration performance of remote sensing image information at various resolutions by jointly extracting local global features [15]. In response to the low efficiency of Convolutional Neural Networks (CNN) in recognizing complex changes, W. Wang et al. designed a RSIC approach on the basis of multi-scale self-attention mechanism and mixed attention module, which improved the detection quality [16].

In addition, to address the sample imbalance in semantic change detection, Q. Zhu et al. built a twin global learning approach, which improved the accuracy and generalization of novel semantic change detection [17]. X. Zhang et al. designed an asymmetric cross attention hierarchical network that combined CNN and self-attention mechanisms to address issues such as poor interaction between CNN and self-attention mechanisms, resulting in significant computational resource consumption [18]. In response to the low accuracy and complex calculation of hyper-spectral remote sensing image detection, H. Firat et al. built a mixed 3D residual space spectral convolutional network to improve the accuracy of hyper-spectral remote sensing image classification [19]. In response to the limitations of convolutional layers on the global capture ability of neural networks, L. Wang et al. built a self-attention mechanism similar to a U-shaped network for urban scene segmentation, which improved the speed and accuracy of RSIC detection [20]. H. Chen et al. built an unsupervised multi-modal change detection framework on the basis of Fourier domain structural relationship analysis to address the difficulties in processing multi-modal remote sensing images caused by modal heterogeneity, which improved the fusion efficiency of local and non-local structural difference maps [21].

In summary, numerous researchers worldwide have noticed the problems in the operation of RSIC detection and have conducted multiple research efforts to address these issues. In addition, accurate and real-time RSIC detection is a prerequisite for expanding the use of remote sensing images at the geographic spatial scale, and its importance is self-evident. However, due to the redundancy of feature extraction information and unreasonable allocation of feature weights in traditional RSIC detection, the detection performance is unstable, and the accuracy and real-time performance of RSIC detection in complex environments are low. Therefore, based on the residual network, combined with attention mechanism and DSC module, a RSIC detection model on the basis of IRAU and DSC is established to improve the feature extraction accuracy and allocate the weights of detection results reasonably. The research aims to provide a comprehensive and innovative solution for addressing the latency and efficiency of RSIC detection in actual natural or artificial environments.

## 3. Methods and materials

With the expansion of industrial scale, the RSIC detection is also facing new challenges. A new RSIC detection model on the basis of IRAU and DSC is developed to address poor accuracy and real-time performance of traditional change detection techniques, which cannot meet diversified detection tasks. The model involves two parts, where IRAU optimizes the detection accuracy, while DSC optimizes the real-time performance of the model, jointly achieving efficient real-time detection of the model.

### 3.1. IRAU unit construction based on accuracy optimization

Accuracy and real-time performance are prerequisites for the application of RSIC detection in many aspects. However, traditional RSIC detection models are often affected by multiple factors, resulting in poor detection accuracy and real-time performance. A RSIC detection model on the basis of IRAU and DSC is designed to address the above issues. Among them, IRAU is responsible for optimizing model accuracy, which includes ResNet variants of -Res2Net＋, SPC, and channel attention. By extracting semantic information from feature maps at various scales, IRAU enriches and refines the

feature information to be detected, ultimately optimizing the accuracy of change detection. The structure of IRAU and Res-2Net+modules is shown in Fig 1.

In Fig 1(a), after the feature image is input into IRAU, it is processed through Res2Net+module, SPC module, and channel attention module to extract fine-grained semantic information of different scales. Finally, it is normalized, multiplied, and overlaid by softmax function to complete the image change detection output. In Fig 1 (b), the $F_{input}$ is input into the Res2Net+module. First, the input is divided into four subsets after the first 1x1 convolution, defined as $X_i$ and $i = 1, 2, 3, 4$, where each feature has the same scale size. In order to reduce parameters while increasing subsets, except for $X_1$, all other sub-features have corresponding 3x3 convolution kernels, defined as $K_{i-1}$, whose output is $Y_i$, as displayed in equation (1).

$$\begin{cases} Y_i = X_i & i = 1 \\ Y_i = K_i(X_i) & i = 2 \\ Y_i = K_i(X_i + Y_{i-1}) & 2 < i \leq 4 \end{cases} \tag{1}$$

In equation (1), $X_1$ is $Y_1$. $Y_2$ is the output of $X_2$ after $K_2$ convolution. When $2 < i \leq 4$ is present, all sub-features $X_i$ are added to $K_{i-1}$ and then input to $K_i$. After convolution, $Y_i$ is output. The channel shuffling operation is added to the Res-2Net+module, and $Y_i$ is concatenated in the channel dimension. Then, the 1*1 convolution is performed to get $F_{CS}$. Finally, $F_{CS}$ and $F_{input}$ are added through residual connections to obtain $F_{res}$, which is beneficial for feature language stacking at different scales [23–24]. In addition, the remaining sub-module structures of IRAU are shown in Fig 2.

As shown in Fig 2 (a), the SPC module takes $F_{res}$ as input and divides $F_{res}$ into four parts, defined as $M_i$. $i = 0, 1, 2, 3$. SPC uses group convolution to process input tensors at different scales. The relationship between the convolution kernel size $S$ and the size of the group convolution integration group $g$ is $g = 2^{\frac{S-1}{2}}$, and $S = 3, 5, 7, 9$. The SPC module obtains a multi-scale feature map $F_i$ through group convolution processing, concatenates $F_i$ to output $F_{spc}$, and independently learns multi-scale spatial information through each segmentation part [27]. In Fig 2 (b), the channel attention module takes $F_{spc}$ as input and generates $F_{avg}^c$ and $F_{max}^c$ feature maps through MaxPool and AvgPool operations. Then, $F_{avg}^c$ and $F_{max}^c$ are forwarded to a Multi-Layer Perceptron (MLP) shared network to obtain a channel attention map $M_c$. The shared network is taken to each post feature map. The output feature vectors are fused through an element wise summation method, as shown in equation (2).

$$M_c(F) = \sigma(MLP(AvgPool(F)) + MLP(maxPool(F))) = \sigma(W_1(W_0(F_{avg}^c)) + W_1(W_0(F_{max}^c))) \tag{2}$$

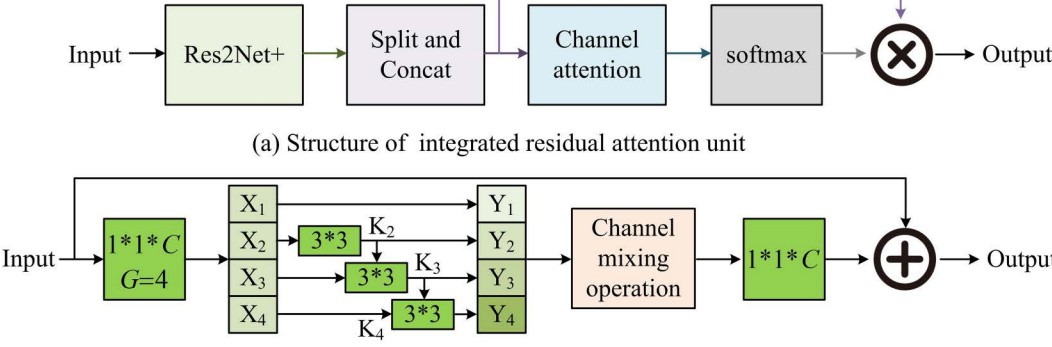

(a) Structure of integrated residual attention unit

(b) Structure of Res2Net+ block

**Fig 1. The structure of IRAU and Res2Net+ block [22].**

Among them, $\sigma$ signifies the sigmoid activation function. $W_0$ and $W_1$ represent the weights generated by MLP, which are shared for both $F_{avg}^c$ and $F_{max}^c$. The sigmoid activation function acts on $W_0$. Based on the weight difference of feature vectors, the channel attention module assigns values to the input image features, which is beneficial for further processing in the future [28]. Finally, the weights $W_0$ and $W_1$ are normalized using IRAU's softmax. The processed feature vector $M_c(F)$ and feature map $F_{spc}$ were pixel multiplied to obtain the output $F_{IRA}$ of IRAU. In summary, the second output $F_{CS}$ of Res-2Net+module, the output $F_{spc}$ of SPC module, and the final output $F_{IRA}$ of IRAU are shown in equation (3).

$$\begin{cases} F_{CS} = \psi\left(\Omega\left([Y_1, Y_2, Y_3, Y_4]\right)\right) \\ F_{SPC} = SPC\left(F_{input} + F_{CS}\right) \\ F_{IRA} = F_{SPC} \otimes \partial\left(CAM\left(F_{SPC}\right)\right) \end{cases} \tag{3}$$

In equation (3), $\psi$ represents a 1*1 convolution operation. [] represents concatenation in the channel dimension. *CAM* signifies channel attention mechanism processing for features. $\otimes$ represents the multiplication operation of feature vector pixels. To further optimize the accuracy of extracting multi-scale features and deep semantic information from RSIC detection models, a Multi-Scale Residual Block (MSRB) is used to extract image features [29]. Its structure is shown in Fig 3.

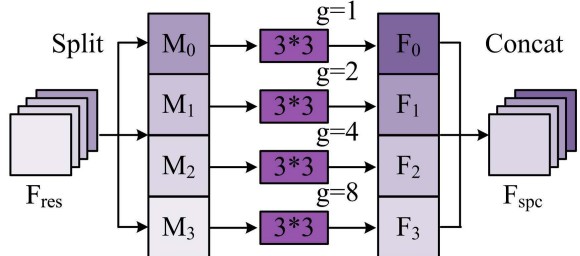

(a) Structure of split and concat block

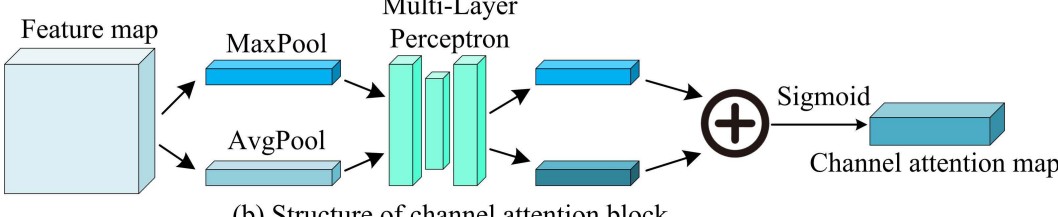

(b) Structure of channel attention block

**Fig 2. The structure of SPC block and channel attention block [25–26].**

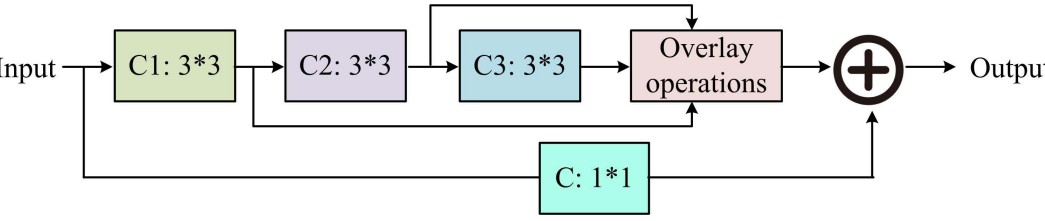

**Fig 3. The structure of multi-scale residual block [30].**

From Fig 3, the study replaces the more complex 5*5 and 7*7 convolution operations by concatenating three 3*3 convolutions to decline computational complexity. After the remote sensing image is input into MSRB, it undergoes three concatenated 3*3 convolution operations before being overlaid. The overlaid output features are then pixel added to the initial output after the 1*1 convolution operation. To capture spatial information in low-level features, a 3*3 convolutional kernel is used. To decline memory consumption, the quantity of filters in the internal convolutional layer is adjusted based on specific parameters, and a 1*1 convolutional kernel is used to ensure the accuracy of feature processing with a 3*3 convolutional kernel. To address the insignificant changes in model features and excessive irrelevant information in traditional detection, an Attention Gate (AG) module is introduced, as shown in Fig 4.

From Fig 4, g signifies the input from the previous module, and x signifies the input from the skip connection. The skip connection is to skip one or several neural layers and transmit information to deeper layers of the neural network [32]. x and g are transformed into the same number of channels through a 1*1 convolution operation. After up-sampling, pixel addition is performed and rectified using a Rectified Linear Unit (ReLU) function. Then, another 1*1 convolution is conducted and the Sigmoid activation function is taken to get the importance score, which is assigned to each part of the feature map. The attention feature map is multiplied with the skip connection input to generate the final output of AG [33]. In summary, to ensure that the proposed IRAU can accurately and efficiently process the features of changing regions, a corresponding image network change detection twin network is set up, as shown in Fig 5.

Fig 5 shows the process of using detection twin networks for detecting changes in images at different temporal phase. This network is based on the Siamese-Difference framework and ensures consistency in feature extraction of dual temporal images through shared weights. Each branch consists of MSRB and 2*2 max pooling layers, where MSRB is applied

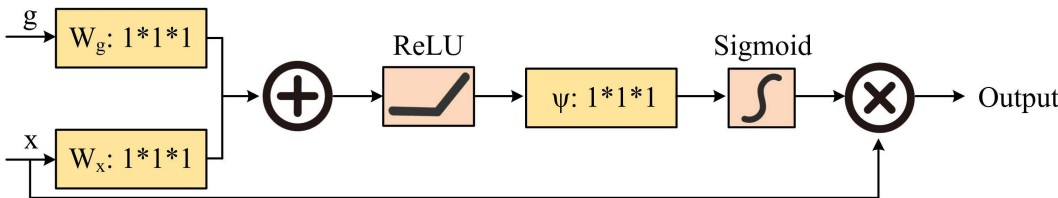

**Fig 4. The structure of attention gate block [31].**

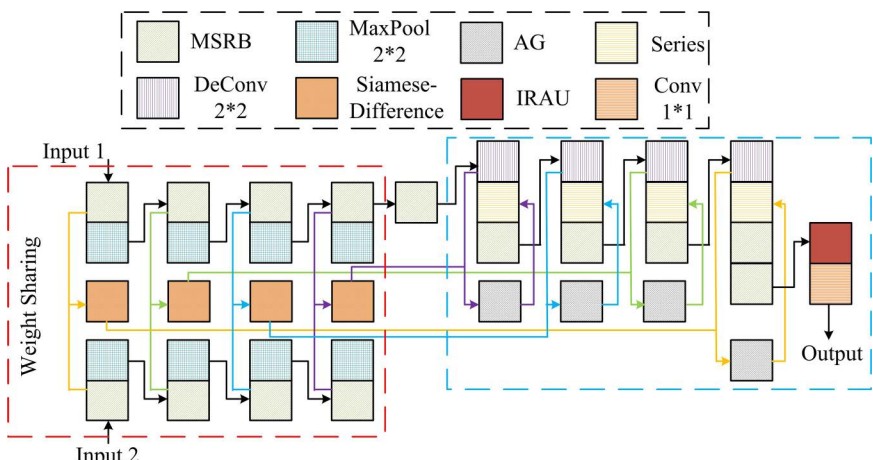

**Fig 5. Detection network structure with IRAU [34].**

to get rich feature information. To optimize the detection accuracy of differences between images of different time phases, the network computes the absolute difference between the MSRB structure streams output by two branches and connects them into a difference feature map. Within the network, the AG module is introduced to combine the down-sampling feature map at the input and the up-sampling feature map at the output, aiming to highlight key changes in the image while suppressing noise. This mechanism emphases the real changing areas, improving the detection performance. The output design includes a combination of 2*2 deconvolution layers and MSRB, with the former responsible for restoring the resolution of the feature map and the latter further refining the features. Next, the feature map is passed to IRAU, which enhances the accuracy of edge recognition in changing regions through detailed multi-scale feature capture and adaptive channel optimization. It is crucial for generating high-quality change maps. Finally, after adjusting the quantity of channels in the feature map through a 1*1 convolutional layer, the change detection result is output.

### 3.2. Construction of dsc module and remote sensing image change detection model based on real-time optimization

The proposed IRAU and its application network have improved the accuracy of the detection model, but there are still challenges to the real-time performance. Due to factors such as climate and environment, traditional RSIC detection models have poor real-time performance and are difficult to complete continuous change detection tasks. Therefore, DSC is introduced into the proposed RSIC detection model, which is responsible for optimizing the real-time change detection performance. To improve the operational efficiency of DSC, the Efficient Channel Attention (ECA) is combined with Enhanced Atrous Spatial Pyramid Pooling (ASPP+) modules to construct an application network suitable for DSC operation. The DSC module and its application network structure are shown in Fig 6.

As shown in Fig 6 (a), DSC divides the convolutional block into two stages: deep convolution and point by point convolution. 3*3 deep convolution is responsible for performing a single filter convolution on each channel of the input image. Then, the batch normalization layer regularization [35] and ReLU activation operations are performed. Afterwards, the output channels of the depth convolution are combined by performing a 1*1 operation point by point convolution. The batch normalization layer regularization and ReLU activation are performed on the obtained feature maps again. In Fig 6 (b), the input of the DSC uses the Siamese-Difference framework to process images of different time phases in the detection area. DSC decline network parameters and computational complexity [36]. Therefore, the study uses DSC instead of conventional convolutional blocks for feature extraction at the input and output ends of its application network. In addition, the study adds ECA before skip connections to effectively integrate the rich spatial information extracted from shallow features at the input end with the rich semantic information extracted from deep features at the output end. Furthermore, to further optimize the detection performance of the overall detection model for targets of different scales in the changing area, an ASPP+module is introduced at the output end. After the feature map undergoes ASPP+operation, a 1*1 convolution is conducted again. Finally, the change map is output. The ECA module structure is shown in Fig 7.

ECA is an improvement based on compression and excitation networks, which uses local cross channel interaction strategies to decline the computational complexity while maintaining detection performance [38]. As shown in Fig 7, Global Average Pooling (GAP) is performed immediately after the feature map is input. Then, the fast one-dimensional convolution is used to perform cross channel fusion on kernel values, where $k = 5$ represents the cross channel range. Afterwards, the channel weight values are obtained through Sigmoid activation operation, enabling the detection model to efficiently learn while avoiding the impact of dimensionality reduction on channel weight updates. ASPP is an algorithm model widely used in remote sensing image segmentation, which captures multi-scale information through dilated convolutions with various dilation rates to avoid detail loss caused by down-sampling. The proposed ASPP+ in the study is an improvement based on ASPP [39], and its structure is shown in Fig 8.

As shown in Fig 8, in ASPP+, the dilated convolution with an original Rate value of 24 is replaced by a 1*1 convolution block. This is because in the traditional ASPP structure, as the Rate parameter of the dilated convolution increases, the

effective area actually involved in the convolution operation in the input image will become smaller and smaller, which may lead to feature information loss and affect the detection performance [40]. After replacing with 1*1 convolutional blocks, the module can maintain a large receptive field while avoiding problems such as the effective convolution area being

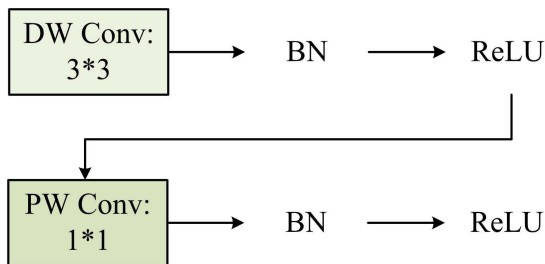

(a) The structure of DSC block

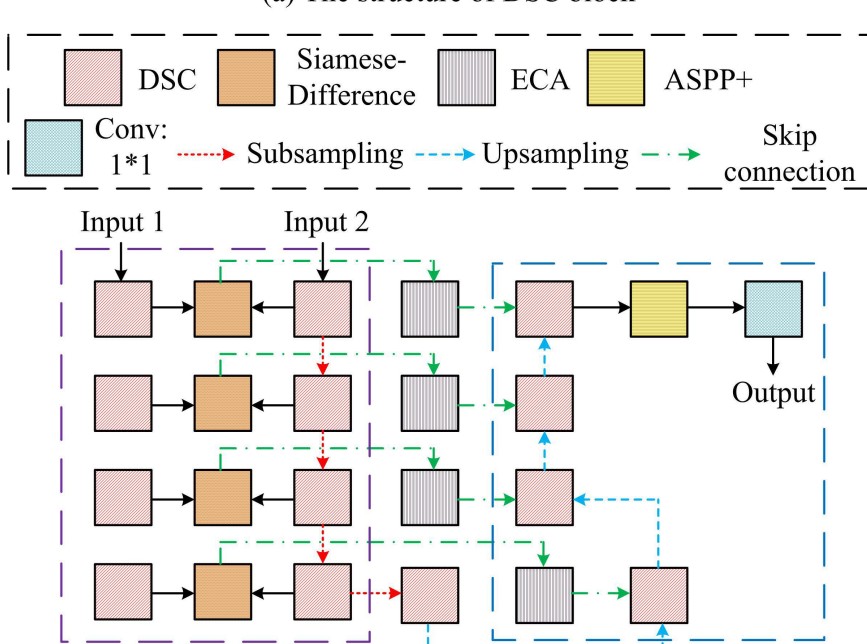

(b) The structure of DSC application network structure

**Fig 6. DSC block and its application network structure.**

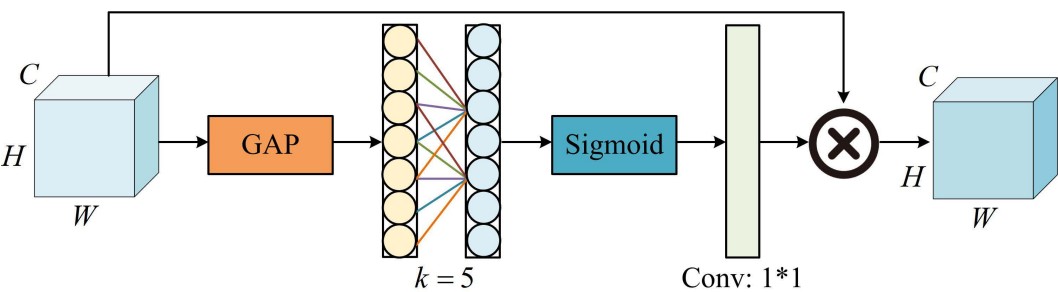

**Fig 7. The structure of efficient channel attention block [37].**

too small. The input feature map is subjected to DSC feature extraction to reduce the computational complexity. Subsequently, a 1*1 convolution is performed in parallel with three other dilated convolutions with different Rate values of 6, 12, and 18, respectively, to capture feature information of various scales in the feature map. Then, the feature maps generated by the above operations are concatenated to obtain a multi-scale feature fusion feature map. Finally, in order to further optimize the feature information, the concatenated feature map undergoes another 1*1 convolution operation, which helps reduce redundant information and achieve efficient fusion of multi-scale features, thereby enhancing the detection capability of the model.

In summary, a RSIC detection network model on the basis of IRAU and DSC is proposed to address the accuracy and real-time issues of change detection models. This model is divided into two parts: IRAU and DSC, aimed at optimizing the detection accuracy and efficiency. The workflow is as follows. Remote sensing images from various time periods are input into two branches of the network, and multi-scale feature extraction and feature map dimensionality reduction are achieved through shared weight MSRB and 2*2 max pooling layers. The feature map is handled by the DSC module to reduce parameters and computational costs. The AG module integrates input down-sampling and output up-sampling feature maps to enhance change features and remove redundant information. The output end first restores the resolution of the feature map through a 2*2 deconvolution layer, and then jumps and connects with the input end feature map to fuse multi-level features. Then, the feature map is further refined by MSRB and sent to the IRAU module for multi-scale feature extraction and adaptive feature refinement. Finally, the feature map is adjusted for the number of channels through a 1*1 convolutional layer to produce a change map. This model effectively improves the accuracy of change detection while declining computational complexity, making it suitable for applications in high-resolution remote sensing images.

## 4. Results

To verify the effectiveness and superiority of the RSIC detection model on the basis of IRAU and DSC, the theoretical basis and algorithm analysis mentioned above are comprehensively studied. The research chooses a deep learning framework on the basis of PyTorch as the experimental platform to conduct model training and simulation testing experiments on different algorithms. In addition, the study also conducts experiments on the change detection performance in practical environments, collects relevant data, analyzes the experimental results in detail to compare the model performance.

### 4.1. Model training and simulation testing experiment

In model training and simulation testing experiments, Windows 10 is chosen as the operating system and AdamW optimizer is used for model training and optimization. The study selects Feature Pyramids within the Network, Deformable Component Detection Model, You Only Look Once, and Deep Image Fusion Network as comparative methods, named

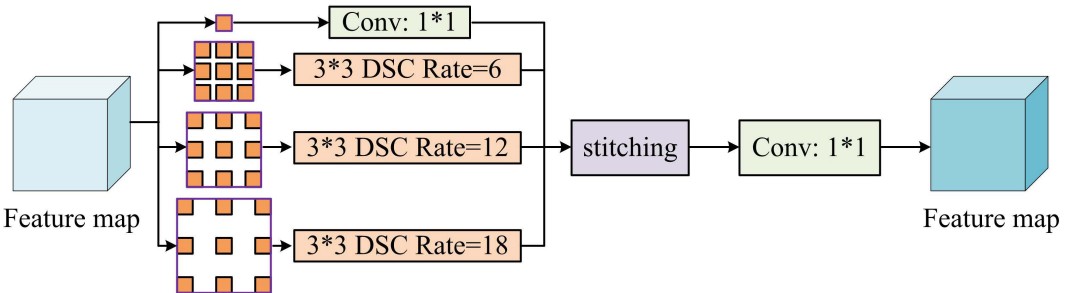

**Fig 8. The structure of ASPP+ block.**

M1, M2, M3, and M4, respectively. The research uses the change detection dataset as a training and test set (8:2) with remotely sensed data from a region in Angers, France (Geographical distribution center: Latitude: 47° 37' 28" N, Longitude: 000° 09' 55" W) covering the period from June 2019 to September 2021, containing Landsat-8 and Sentinel-2 multi-temporal imagery with a resolution of 10–30 meters. The data sources are the U.S. Geological Survey and European Space Agency public databases, covering typical scenarios such as urban sprawl and vegetation changes. The study first trains the model to verify its training efficiency, as presented in Fig 9.

According to Fig 9 (a), IRAU-DSC completed convergence at 1700 iterations, with the fastest convergence speed, followed by M1, which completed convergence at 2200 iterations. M3 and M4 converged at 2400 and 3000 iterations, respectively, while M2 had the slowest convergence speed and completed it at 3700 iterations. From Fig 9 (b), before and after convergence, M2 had the lowest average accuracy, at 0.25 and 0.68, respectively, followed by M4, with average accuracy before and after convergence of 0.36 and 0.83, respectively. The accuracy of M1 and M3 before and after convergence was between 0.36–0.75 and 0.87–0.96, respectively, while IRAU-DSC had the highest accuracy before and after convergence, with mean values of 0.54 and 0.97. The IRAU-DSC has better learning performance than other methods and the highest accuracy. Afterwards, to verify the stability of the trained algorithm in detecting images, repeated recognition experiments are conducted, as presented in Fig 10.

According to Fig 10 (a), the detection accuracy of M2 varied the most, ranging from 67.53% to 76.94%, followed by M4, with accuracy fluctuating between 83.12% and 88.93%. The detection accuracy of M1 and M3 showed relatively small changes, with maximum amplitudes of 2.68% and 5.81%, respectively, while the accuracy of IRAU-DSC showed the smallest changes, ranging from 95.82% to 99.68%. According to Fig 10 (a), in terms of average accuracy, IRAU-DSC had the highest accuracy, reaching 97.56%. Next was M1, with an average accuracy of 94.68%. The average accuracy of M3 and M4 was slightly lower, at 90.91% and 85.91% respectively, while M2 had the lowest accuracy, with an average of only 71.97%. After training on the dataset, the performance of IRAU-DSC is more stable, and the accuracy is consistent with the above training results. Subsequently, the study conducts Receiver Operating Characteristic curve (ROC) and Area Under Curve (AUC) analysis, as shown in Fig 11.

From Fig 11 (a) and (b), in the ROC, the false positive rate increased the fastest when it was between 0–0.3, and the true positive rate slowed down when it was between 0.3–0.6, and gradually stabilized after 0.6. When the false positive rate was the same, M2 had the lowest true positive rate, followed by M4. At this point, the true positive rates of M1 and M3 were slightly higher than M4, but IRAU-DSC had the highest true positive rate. In addition, there was a similar trend in

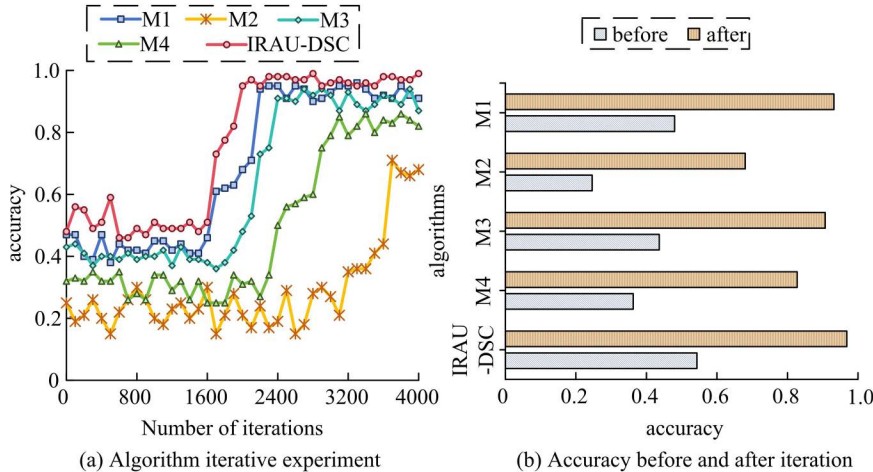

(a) Algorithm iterative experiment (b) Accuracy before and after iteration

**Fig 9. Algorithm training accuracy experiments.**

the AUC values of different algorithm models. The AUC value of IRAU-DSC was the highest, reaching 0.90, followed by M1 with an AUC of 0.85. The AUC of M3 and M4 was 0.82 and 0.77, with M2 having the lowest AUC value of only 0.70. The IRAU-DSC has higher change detection performance and fully meets the needs of remote sensing detection tasks. And Kappa coefficient can reflect the degree of consistency between the classification results and the real labeling, while excluding the consistency caused by random factors. Therefore, the study compares the Kappa coefficients of different methods to verify their accuracy for change detection maps, and the results are shown in Table 1.

In Table 1, the Kappa coefficient of IRAU-DSC reaches 0.90, which is significantly higher than that of M1 (0.75), M2 (0.78), M3 (0.72) and M4 (0.82), and its TP (800) and TN (180) are optimal and FP (20) and FN (0) are the lowest, which verifies that the model has a high accuracy and anti-jamming ability in the recognition of changing regions. Compared with

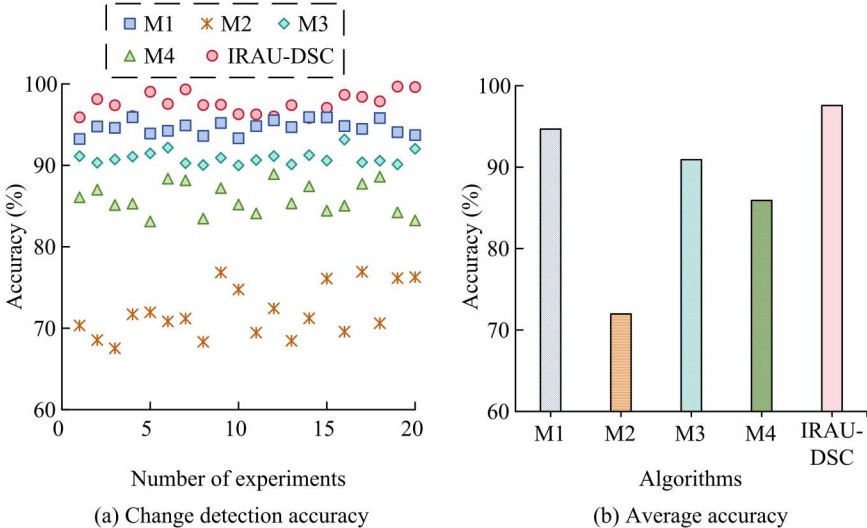

(a) Change detection accuracy  (b) Average accuracy

**Fig 10. Repeat the detection experiment.**

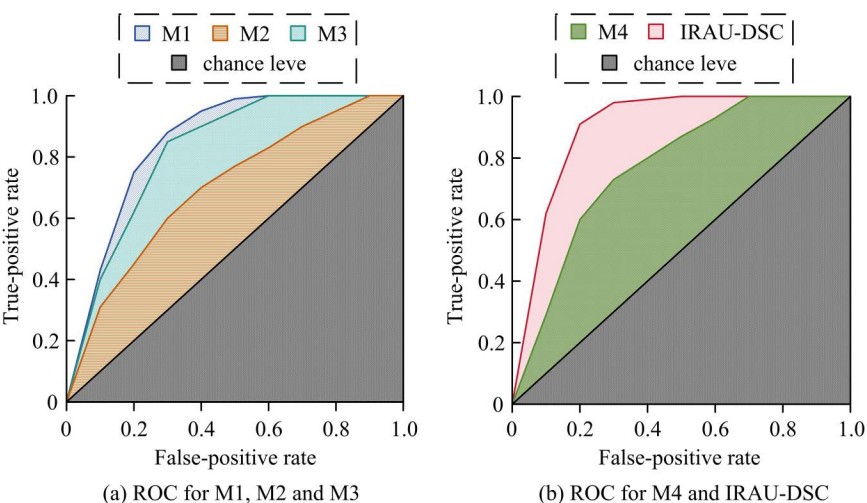

(a) ROC for M1, M2 and M3  (b) ROC for M4 and IRAU-DSC

**Fig 11. Receiver operating characteristic curves for different algorithms.**

**Table 1. Kappa coefficients for different algorithms.**

| Methods | TP | TN | FP | FN | Kappa Coefficient |
|---|---|---|---|---|---|
| M1 | 700 | 200 | 50 | 50 | 0.75 |
| M2 | 720 | 180 | 70 | 30 | 0.78 |
| M3 | 680 | 220 | 30 | 70 | 0.72 |
| M4 | 750 | 150 | 100 | 0 | 0.82 |
| IRAU-DSC | 800 | 180 | 20 | 0 | 0.9 |

**Table 2. Ablation experiments of various block.**

| algorithms | Block | | | | | | Accuracy (%) | Time Deviation(ms) |
|---|---|---|---|---|---|---|---|---|
| | MSRB | AG | IRAU | ECA | ASPP+ | DSC | | |
| M1 | √ | × | × | × | × | × | 93.25 | 1775 |
| M2 | × | × | × | × | × | × | 70.34 | 2846 |
| M3 | × | × | × | × | × | × | 91.13 | 2169 |
| M4 | × | × | × | × | × | × | 86.08 | 2198 |
| IRAU-DSC | √ | √ | × | × | × | × | 92.19 | 1130 |
| IRAU-DSC | √ | √ | √ | × | × | × | 96.07 | 1129 |
| IRAU-DSC | × | × | × | √ | √ | × | 91.25 | 471 |
| IRAU-DSC | × | × | × | √ | √ | √ | 94.16 | 236 |
| IRAU-DSC | √ | √ | × | √ | √ | × | 96.51 | 299 |
| IRAU-DSC | √ | √ | √ | √ | √ | √ | 98.42 | 182 |

traditional methods such as in-network feature pyramid and deformable part detection, the proposed algorithm effectively improves the robustness and discriminative consistency of change detection in complex scenes through deep feature fusion strategy. Experimental results show that the accuracy of IRAU-DSC for change detection graph is much higher than other methods. To further validate the importance of the proposed IRAU and DSC models in detection tasks, ablation experiments are conducted on the detection models, as displayed in Table 2.

According to Table 2, the detection accuracy of M1, M2, M3, and M4 ranged from 70.34% to 93.25%, with the latency of 1775ms, 2846ms, 2169ms, and 2198ms, respectively. At this time, the detection accuracy of IRAU-DSC was 98.42%, with the latency of 182ms. When the detection model proposed in the study only had MSRB and AG, its accuracy was 92.19% and the latency was 1130ms. After introducing IRAU, its accuracy increased by 3.88% and the latency remained almost unchanged. When the detection model only had ECA and ASPP+, its accuracy was 91.25% and the latency was 471ms. After introducing DSC, the accuracy increased by 2.91% and the latency decreased by 235ms. In addition, when IRAU-DSC lost IRAU and DSC, its accuracy decreased by 1.91% and latency increased by 117ms. The introduced module exerts a crucial role in the overall detection model, and the proposed improvements to IRAU and DSC are the core for achieving excellent performance of the model.

### 4.2. Experimental study on the performance of model actual change detection

Based on the above results, the study compares M1 and M2 algorithms, with IRAU-DSC as the research object. The detection performance in the time environment is analyzed to verify its potential for practical application and promotional value. The study first selects houses, rivers, and roads, and compares their actual performance through different algorithms to detect different elements. The results are shown in Fig 12.

As shown in Fig 12 (a), M1 had a higher overall detection efficiency than M2, but lower than IRAU-DSC. M1 had the highest detection efficiency for rivers, at 0.88, and for houses and roads, at 0.83 and 0.86 respectively. As shown in Fig

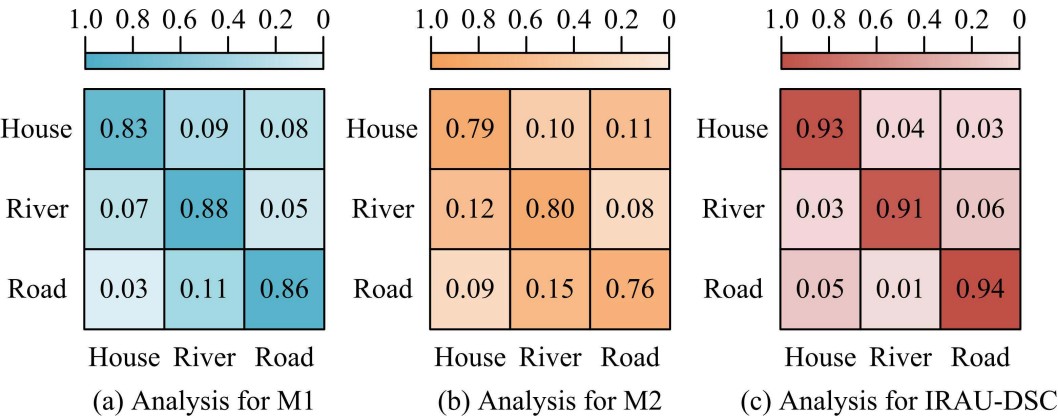

Fig 12. Confusion matrix analysis for different algorithms.

12 (b), M2 had the lowest detection efficiency in practical applications, ranging from 0.76 to 0.80, and M2 also had good detection efficiency for rivers. According to Fig 12 (c), IRAU-DSC had high detection performance for houses, rivers, and roads, with an overall efficiency of over 0.91. Among them, the algorithm had the highest detection efficiency for roads, reaching 0.94. The experimental results show that in practical applications, IRAU-DSC has better detection efficiency for different elements than other algorithms and can perform remote sensing detection in complex environments. Then, to verify the performance in detecting changes at the spatial scale, the study selects images of a large courtyard at different time phases for change detection experiments, as presented in Fig 13.

From Figs 13 (a) and (b), the main change in this courtyard was the dismantling of the tool room in the upper right corner, which was converted into a sun drying area. There were more temporary material storage areas in the middle, and larger warehouses were developed on the barren land in the lower part. In addition, there were significant changes in the number of cars in two different temporal images. From Fig 13 (c), IRAU-DSC detected changes in the tool room and drying area in the upper right corner, with a slight deviation in the detection of changes in the temporary stacking area in the middle. It performed well in detecting changes in the lower warehouse and detected 17 changes in vehicles. From Fig 13 (d), M1 did not detect the dismantling of the tool room, and the temporary material stacking area and warehouse in the middle and lower parts were found to be distorted. Nine changes in the vehicles were detected, but some of the detections showed deviations. From Fig 13 (e), M2 experienced severe distortion when detecting tool room, drying area, and warehouse, and did not detect any changes in the temporary stacking area. M2 also detected changes in 7 vehicles, but there were significant deviations. To further verify the real-time detection performance on a continuous time scale, the study performs continuous change detection on a white car driving at a certain intersection. The results are shown in Fig 14.

From Fig 14 (a), at the 3rd second of the screen, the car entered the screen and traveled along the right side. At this time, IRAU-DSC and M1 detected the position of the target car, but M1 showed a significant deviation, while M2 erroneously detected the gray car on the right side. From Fig 14 (b), at the 11th second of the screen, when the car reached the upper position of the screen, IRAU-DSC still correctly detected the target position, while the deviation of M1 increased and the detection latency of M2 was larger. The detection result was the target vehicle position around the 10th second. From Fig 14 (c), at the 17th second of the screen, when the car reached the left edge of the screen, there was a slight deviation in IRAU-DSC detection. The detection deviation of M1 was further increased, and the detection result of M2 still had a latency, failing to correctly detect the position of the current target car. The IRAU-DSC has high real-time and accuracy in detecting continuous changes of moving targets. To verify the authenticity of the actual detection efficiency experiment mentioned above, the accuracy and latency time of algorithm detection in practical application scenarios are calculated, as presented in Table 3.

According to Table 3, in practical application scenarios, the detection efficiency decreased. M2 had the lowest accuracy, ranging from 65.46% to 80.49%, and the highest latency, ranging from 2435ms to 2966ms. M1 had slightly higher efficiency, with accuracy and latency ranging from 83.67% to 92.65% and 1194ms-1722ms, respectively. The accuracy

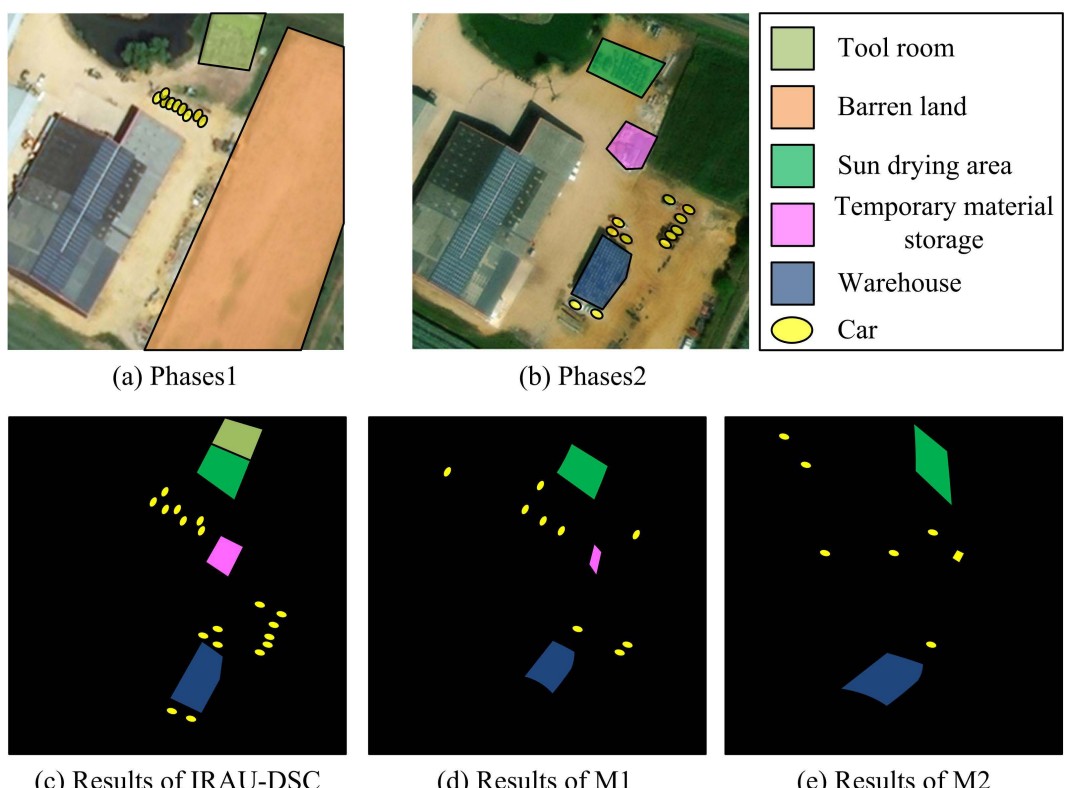

(a) Phases1 (b) Phases2

Legend:
- Tool room
- Barren land
- Sun drying area
- Temporary material storage
- Warehouse
- Car

(c) Results of IRAU-DSC (d) Results of M1 (e) Results of M2

**Fig 13. Change detection on a spatial scale.** (Source from [https://earthexplorer.usgs.gov/](https://earthexplorer.usgs.gov/); Latitude: 47° 37' 28" N, Longitude: 000° 09' 55" W).

(a) The third second (b) The eleventh second (c) The seventeenth second

**Fig 14. Change detection on a continuous time scale. (Source from: [https://pixabay.com/zh/videos/cars-roundabout-rondo-flowers-road-168811/](https://pixabay.com/zh/videos/cars-roundabout-rondo-flowers-road-168811/)).**

and latency of IRAU-DSC were still superior to other algorithms, with accuracy ranging from 90.27% to 93.94%, with an average of 92.43%, and latency ranging from 202ms to 318ms, with an average of 260ms. The experimental results show that in practical environmental applications, the detection efficiency calculation data of IRAU-DSC is consistent with the detection results, and the experimental results are true and reliable. The excellent performance of IRAU-DSC comes from the proposed IRAU units and the optimization improvement of IRAU and DSC. In addition, in order to verify the generalization and robustness of the research model in different spectral scenarios, the study selects different spectral scenarios for change detection experiments, and the results are shown in Table 4.

In Table 4, the study validates the model performance in hyperspectral (224 bands), multispectral (8 bands), and low-resolution multispectral (4 bands) scenarios.IRAU-DSC achieves the optimal Accuracy (93.5%) and Kappa (0.90) in hyperspectral scenarios, with an improvement of 6.8%/4.8% and 0.09/0.07 over M1/M2, and a reduction of 16 ms to 12 ms in the temporal bias (28.7 ms). It maintains 85.4% accuracy and 0.81 Kappa for low-resolution data, and improves 5.2%−7.2% accuracy and 0.08–0.10 Kappa compared with the baseline model, and the computational efficiency is stable (19.5ms), which indicates that its adaptive spectral feature fusion mechanism can effectively deal with the band difference and noise interference, and can balance the accuracy and real-time performance. It shows that its adaptive spectral feature fusion mechanism can effectively cope with band differences and noise interference, and take into account both accuracy and real-time performance. Finally, in order to verify the consistency of the model detection results with the ground data, the study selects the Intersection over Union (IoU), the ratio of correctly categorized pixels (changed/unchanged) to

**Table 3. The actual change detection efficiency of algorithms.**

| Number of experiments | Accuracy(%) | | | Time deviation(ms) | | |
|---|---|---|---|---|---|---|
| | M1 | M2 | IRAU-DSC | M1 | M2 | IRAU-DSC |
| 1 | 84.47 | 68.73 | 93.34 | 1535 | 2574 | 211 |
| 2 | 87.95 | 70.78 | 92.00 | 1722 | 2768 | 207 |
| 3 | 85.34 | 68.25 | 93.94 | 1469 | 2499 | 302 |
| 4 | 86.96 | 65.46 | 91.93 | 1347 | 2559 | 281 |
| 5 | 85.79 | 80.49 | 90.27 | 1360 | 2966 | 318 |
| 6 | 83.67 | 70.74 | 90.56 | 1279 | 2435 | 253 |
| 7 | 84.77 | 66.11 | 93.68 | 1194 | 2706 | 296 |
| 8 | 92.65 | 67.05 | 92.72 | 1370 | 2722 | 202 |
| 9 | 85.74 | 70.58 | 92.38 | 1427 | 2578 | 257 |
| 10 | 85.58 | 70.18 | 93.43 | 1610 | 2743 | 273 |
| Mean | 86.29 | 69.84 | 92.43 | 1431 | 2655 | 260 |

**Table 4. Comparison of model performance in different different spectral scenarios.**

| Scene conditions | Methods | Accuracy (%) | Time Deviation (ms) | Kappa Coefficient |
|---|---|---|---|---|
| Hyperspectral (224 bands) | M1 | 86.2 | 34.5 | 0.81 |
| | M2 | 88.7 | 41.2 | 0.83 |
| | IRAU-DSC | 93.5 | 28.7 | 0.9 |
| Multi-spectral (8 bands) | M1 | 82.4 | 28.1 | 0.76 |
| | M2 | 84.9 | 33.6 | 0.79 |
| | IRAU-DSC | 89.3 | 24.9 | 0.86 |
| Low resolution multispectral (4 bands) | M1 | 78.6 | 21.3 | 0.71 |
| | M2 | 80.2 | 25.8 | 0.73 |
| | IRAU-DSC | 85.4 | 19.5 | 0.81 |

the total pixels (Correct Pixel Ratio), and the proportion of undetected in the real change area (Omission Error), Position Error and Area Error are compared, and the results are shown in Table 5.

In Table 5, the study verifies the consistency of the detection results with the ground data through IoU, CPR and other indicators. Among them, the IoU of IRAU-DSC reaches 82.6%, which is 10.3% and 6.8% higher than that of M1 (72.3%) and M2 (75.8%); and its Correct pixel ratio (93.4%) is significantly higher than that of the baseline model (86.1%/88.7%). Its Correct pixel ratio (93.4%) is significantly higher than that of the baseline model (86.1%/88.4%). Moreover, the omission error (8.3%) and location error (2.1 pixels) of IRAU-DSC are reduced by 6.2%−9.4% and 1.8 pixels-2.6 pixels, respectively, and the area error (5.3%) is reduced by 7.5%/4.3% compared with that of the M1/M2 model, which proves that its detection boundary, spatial location, and area coverage are highly matched with the real area of the change, which effectively guarantees the change detection results. It effectively guarantees the ground verifiability of the change detection results.

## 5. Discussion and conclusion

In response to the low accuracy and poor real-time performance of traditional remote sensing change detection models, this study proposed the IRAU unit and improved it by combining modules such as DSC. Finally, the IRAU-DSC model was proposed, which improved the detection performance by increasing the receptive field and reasonable weighting values. During training on the dataset, the average accuracy of IRAU-DSC before and after convergence was 0.54 and 0.97, respectively, while the accuracy ranges of other algorithms before and after convergence were 0.15–0.75 and 0.66–0.96, respectively. After training, the repeat detection accuracy of IRAU-DSC ranged from 95.82% to 99.68%, while the average accuracy of other algorithms were 94.68%, 71.97%, 90.91%, and 85.91%, respectively. In ROC analysis, the AUC value of IRAU-DSC was 0.90, while the AUC values of other algorithms were between 0.70 and 0.85. In the ablation experiment, after losing IRAU and DSC, the accuracy of the model decreased by 1.91% and the latency increased by 117ms. In addition, in actual detection, the detection efficiency of IRAU-DSC was above 0.91, while the efficiency of other algorithms was between 0.76–0.88. In spatial scale detection experiments, IRAU-DSC was more accurate in detecting changes in size within space, while other algorithms suffered from low detection accuracy, distortion, or deviation. In continuous detection experiments, IRAU-DSC showed high accuracy and real-time performance, while other algorithms encountered issues such as error detection, drift, and latency. In the actual detection efficiency calculation, the average latency time of IRAU-DSC was 260ms, while the latency time of other algorithms was between 1194ms-2966ms. In summary, the research has practical application value in improving the accuracy and real-time performance of RSIC detection. However, IRAU relies on multiple modules connected in series with jump connections, which may lead to gradient flow redundancy and reduced feature fusion efficiency. And although DSC reduces the computational volume, the sensory field is limited in high-resolution input, and the fine-grained feature extraction is insufficient. To address the above problems, the research will simplify the module interaction path to optimize the gradient propagation efficiency, design adaptive cavity convolution to enhance the receptive field of DSC, introduce a cross-module dynamic weight allocation mechanism to improve the compatibility of heterogeneous features, and explore the lightweight architecture to balance the computational load and detection accuracy.

**Table 5. Consistency verification of model detection results with ground data.**

| Metric | IoU (%) | Correct pixel ratio (%) | Omission error (%) | Position error (pixel) | Area error (%) |
|---|---|---|---|---|---|
| M1 | 72.3 | 86.1 | 18.5 | 4.7 | 12.8 |
| M2 | 75.8 | 88.7 | 14.2 | 3.9 | 9.6 |
| IRAU-DSC | 82.6 | 93.4 | 8.3 | 2.1 | 5.3 |

## Supporting information

**S1 File. Minimal data set definition.**
(DOC)

## Author contributions

**Methodology:** Yingying Liu.

**Supervision:** Yingying Liu.

**Writing – original draft:** Yingying Liu.

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
